# An Application of the Orthogonal Matching Pursuit Algorithm in Space-Time Adaptive Processing

**DOI:** 10.3390/s20123468

**Published:** 2020-06-19

**Authors:** Anna Ślesicka, Adam Kawalec

**Affiliations:** Faculty of Electronics, Institute of Radioelectronics, Military University of Technology, Gen. S. Kaliskiego 2, 00-908 Warsaw, Poland; anna.slesicka@wat.edu.pl

**Keywords:** space-time adaptive processing, STAP, sparse representation, MIMO radar, orthogonal matching pursuit algorithm, clutter covariance matrix

## Abstract

The article presents a new space-time adaptive processing (STAP) method for target detection in a heterogeneous and non-stationary environment. In study it was proven that it is possible to estimate the clutter covariance matrix (CCM) in STAP by using the MIMO (Multiple Input Multiple Output) radar geometry model and the orthogonal matching pursuit (OMP) algorithm. For the estimation of spatio-temporal spectrum of clutter and target, a model of joint sparse recovery was established. As a result, clutter suppression and target detection in a heterogeneous environment will be achieved. In addition, the proposed method uses a single snapshot of the radar data cube, which eliminates the need for access to all training cells.

## 1. Introduction

STAP is an effective method for airborne radar system to suppress clutter and detect targets. Currently, many scientists studying the possibility of using the STAP technique are trying to answer the question of how to accurately estimate the key in the entire STAP processing algorithm, the clutter covariance matrix (CCM).

Classic, statistical STAP algorithms suppress clutter by estimating the CCM, which is based on the data contained in the training cells surrounding the range cell under scrutiny. Unfortunately, the clutter occurring in a real environment is heterogeneous. Hence, the data contained in the training cells do not reflect the statistical properties of clutter. In addition, such algorithms require a large number of independent and identically distributed training cells, which is also difficult to fulfil [1,2,3].

In connection with the above, a lot of research is conducted to develop a method for estimating the CCM in a heterogeneous environment. A serious of effective methods have been proposed, mainly including direct data domain (D3) methods [4,5,6], the compensation methods of non-side looking geometry [7,8], knowledge-aided (KA) methods [9,10] and the sparse recovery (SR) methods [11,12,13,14,15,16,17].

Currently, the most advanced attempts are carried out to obtain the method of estimating the CCM by the use of a sparse recovery method with prior target information. It has been demonstrated by using a small number of training cells or in some cases one and by using the sparsity of clutter in the spatio-temporal domain that high resolution of the spatio-temporal clutter spectrum can be obtained followed by accurate estimation of the CCM.

In [12,13], sparse recovery methods are proposed for estimating the CCM, which directly uses the data contained in the test cell. As a result of this work, the accuracy of the estimation of the CCM was improved, and the performance in terms of clutter suppression and target detection was enhanced.

This article proved that it is possible to estimate the CCM by using the MIMO radar geometry model and the OMP algorithm. This configuration in combination with the properly selected system and environmental parameters allowed us to obtain very good results, and helped us to understand the approach. As a result, clutter suppression and target detection in a heterogeneous environment will be achieved. In addition, the proposed method uses a single snapshot of the radar data cube, which eliminates the need for access to all training cells.

The spatio-temporal clutter spectrum estimation problem is expressed as the joint sparse recovery problem under a sparse complete basis. The OMP algorithm was used to estimate the spatio-temporal clutter spectrum. The use of the OMP algorithm seems to be easier in practice than other methods; e.g., M-FOCUSS (M-FOCal Underdetermined System Solver). Hence, this solution was adopted as optimal. Finally, the estimated CCM is used to determine the weight vector, which causes clutter suppression and target detection. Compared with the existing methods, the proposed method allows for a more accurate estimation of the CCM and better performance of clutter suppression, as evidenced by the experiments and numerical calculations.

## 2. Model of System Geometry and Model of Signal

A MIMO radar with a linear uniformly array (ULA) mounted on an aircraft flying at altitude *H* with a constant speed of *V* is considered. In Figure 1 is shown the considered geometry of the MIMO radar system. The system consists of *N* receivers spaced apart by *d_R_* and *M* transmitters spaced apart by *d_T_* = *αd_R_*, where *α* is a specific factor. In each transmitter, *K* pulses are transmitted with a pulse repetition frequency of *f_R_*. *𝜃_p_* is the angle between the antenna array and the direction of flight of the platform. It was assumed that the platform velocity vector is perfectly aligned with the antenna array axis vector; hence, *𝜃_p_* = 0. It was assumed that the transmitted signals from different transmitters are independent and coherent.

Considering that the clutter echo data of the range test cell are the superpositions of the echoes of multiple discrete clutter patches on the range cell, the normalized Doppler frequency and the normalized spatial frequency of the ith clutter patch are expressed as [18]:(1)$$f_{di}=\frac{2V}{\lambda f_{R}}\cos\left(\varphi_{i}\right)\cos\left(\theta_{i}-\theta_{p}\right),$$
(2)$$f_{si}=\frac{d_{R}}{\lambda }\cos\left(\varphi_{i}\right)\cos\left(\theta_{i}\right),$$
where *λ* is the wavelength, *φ_i_* is the elevation angle and *θ_i_* is the azimuth angle. Doppler frequency is related to the relative velocity relationship between a target or individual clutter patches. Spatial frequency shows the phenomenon of time difference between the arrival of signals from target or individual clutter patches to individual radar system antennas. Thus, the echo signal received by the nth element of the array corresponding to the mth transmitter and the kth pulse can be represented as [18]:(3)$$x_{m,n,k}=I_{m,n,k}+T_{m,n,k}+n_{m,n,k},$$
(4)$$I_{m,n,k}=\sum_{i=1}^{Nc}\delta_{i}e^{j2\pi \left[\left(\alpha \left(m-1\right)+\left(n-1\right)\right)f_{si}+\left(k-1\right)f_{di}\right]},$$
(5)$$T_{m,n,k}=\delta_{t}e^{j2\pi \left[\left(\alpha \left(m-1\right)+\left(n-1\right)\right)f_{st}+\left(k-1\right)f_{dt}\right]},$$
(6)$$\alpha =\frac{d_{T}}{d_{R}},$$
where *i* = 1, 2, …, N_C_ denotes the number of discrete clutter patches; *f_dt_* and *f_st_* are the normalized Doppler frequency and normalized spatial frequency of the target; *α* denotes the ratio of the distance between the transmitting antennas and the distance between the receiving antennas; *δ_i_* is the reflection coefficient of the ith clutter patch; *δ_t_* is the reflection coefficient of the target; and *n_m,n,k_* denotes noise.

By collating the received echo of the mth transmitted waveform for all receivers and pulses, it was received as
(7)$$x_{m}={\left[x_{m,1,1},x_{m,2,1}\ldots ,x_{m,N,K}\right]}^{T}=I_{m}+T_{m}+n_{m},$$
(8)$$I_{m}=\sum_{i=1}^{N_{c}}\beta_{i,m}\;S\left(f_{di},f_{si}\right),\;$$
(9)$$T_{m}=\beta_{t,m}S\left(f_{dt},f_{st}\right),\;$$
(10)$$\beta_{t,m}=e^{\left[j2\pi \alpha \left(m-1\right)f_{st}\right]},\;$$
(11)$$\beta_{i,m}=e^{\left[j2\pi \alpha \left(m-1\right)f_{si}\right]},\;$$
where ***β_t,m_*** is the reflection coefficient of the target corresponding to the mth transmitted signal; ***β_i,m_*** is the reflection coefficient of the ith clutter patch corresponding to the mth transmitted signal; and ***n****_m_* is a received noise. ***S***(*f_di_, f_si_*) and ***S***(*f_dt_, f_st_*) are the space-time steering vector of the ith clutter patch and target, which can be represented as [18]:(12)$$S\left(f_{di},f_{si}\right)=\left[\begin{array}{c} 1\\ \vdots \\ e^{\left(j2\pi f_{di}\left(K-1\right)\right)} \end{array}\right]⨂\left[\begin{array}{c} 1\\ \vdots \\ e^{\left(j2\pi f_{si}\left(N-1\right)\right)} \end{array}\right],$$
(13)$$S\left(f_{dt},f_{st}\right)=\left[\begin{array}{c} 1\\ \vdots \\ e^{\left(j2\pi f_{dt}\left(K-1\right)\right)} \end{array}\right]⨂\left[\begin{array}{c} 1\\ \vdots \\ e^{\left(j2\pi f_{st}\left(N-1\right)\right)} \end{array}\right],$$
where ⨂ denotes the Kronecker product.

## 3. Joint Sparse Recovery Model

For the estimation of spatio-temporal spectrum of clutter and target, a model of joint sparse recovery was established. The problem of spatio-temporal spectrum estimation was expressed as a problem of optimization of joint sparse recovery based on a complete basis of steering vector.

Due to the above, the plane of normalized Doppler frequency and normalized spatial frequency were divided into a grid with the dimensions *K_d_* × *N_s_*, so as to obtain denser coverage of the analysed range cell. To perform high-resolution spectrum estimation, *K_d_* and *N_s_* values should satisfy the dependence *K_d_N_s_* >> *NK*.

Therefore, the data received by the radar corresponding to a specific range cell can be expressed as [19]
(14)$$x_{m}=\Psi \gamma_{m}+n_{m},$$
where ***γ**_m_* is the clutter and target spatio-temporal spectrum of the range cell under tests and the space-time sparsifying dictionary can be constructed as
(15)$$\Psi =\left[S\left(f_{d1},f_{s1}\right),\ldots ,S\left(f_{dp},f_{sq}\right),\ldots ,S\left(f_{dK_{d}},f_{sN_{s}}\right)\right],$$
where *p* = 1, 2, …, *K_d_*, *q* = 1, 2, …, N_s_. ***S***(*f_dp_, f_sq_*) denotes the space-time steering vector of the (*p*–*q*)-th Doppler and spatial frequency pair:(16)$$S\left(f_{dp},f_{sq}\right)=\left[\begin{array}{c} 1\\ \vdots \\ e^{\left(j2\pi f_{dp}\left(K-1\right)\right)} \end{array}\right]⨂\left[\begin{array}{c} 1\\ \vdots \\ e^{\left(j2\pi f_{sq}\left(N-1\right)\right)} \end{array}\right],$$

To analyse the sparsity of ***γ****_m_*, Equation (14) can be written as [19]:(17)$$x_{m}=\Phi B_{m}+\beta_{t,m}S\left(f_{dt},f_{st}\right)+n_{m},$$
where
(18)$$\Phi =\left[S\left(f_{d1},f_{s1}\right),S\left(f_{d2},f_{s2}\right),\ldots ,S\left(f_{dN_{c}},f_{sN_{c}}\right)\right],$$
(19)$$B_{m}=\left[\begin{array}{c} \beta_{1,m}\\ \begin{array}{c} \beta_{2,m}\\ \vdots \end{array}\\ \beta_{N_{c},m} \end{array}\right],$$

According to Brennan’s rule, the rank of the clutter covariance matrix (*R_c_*) is a measure of the minimum number of adaptive degrees of freedom necessary for a STAP processor:(20)$$rank\left(R_{c}\right)=N+\left(M-1\right)\frac{2V}{d_{R}f_{R}},$$

According to Ward [17], where the Brennan rule regarding the system’s degrees of freedom, including clutter, has been described and analyzed, it can be concluded that the clutter can be represented by the space-time steering vectors, which are spanned by *Q* = rank(*𝚽*) clutter subspace in *𝚽*. Therefore, Equation (17) can be rewritten as:(21)$$x_{m}=V_{m}\Xi_{m}+\beta_{t,m}S\left(f_{dt},f_{st}\right)+n_{m},$$
where ***V**_m_* is a matrix constructed by the space-time steering vectors selected from the matrix *𝚽* and ***𝛯**_m_* is the corresponding reflection coefficient vector:(22)$$\Xi_{m}={\left[\sigma_{m,1},\sigma_{m,2},\ldots ,\sigma_{m,Q}\right]}^{T},$$

Equation (21) indicates that the received data ***x****_m_* of the mth transmitted signal can be represented by space-time steering vectors covering the clutter subspace and the target. Thus, the spectrum ***γ****_m_* can be expressed by steering vectors from the dictionary ***𝛹***. Accordingly, the ***γ****_m_* spectrum can be obtained by solving the following optimization problem [18]:(23)$$\min\|\gamma_{m}\|{}_{0},$$
(24)$$s.t.\;\|x_{m}-\Psi \gamma_{m}\|{}_{2}\leq \varepsilon ,$$
where ‖**·** ‖_u_ denotes the u-norm of matrix or vector; *ε* is a constant determined by noise; and s.t. denotes such that. As shown in article [18], for any transmitted signal *m_*_* (*m_*_* ≠ *m*), echo data can be expressed as
(25)$$x_{m_{*}^{\prime }}=\Phi DB_{m^{\prime }}+e^{\left[j2\pi \alpha \left(m_{*}^{\;}-m\right)f_{st}\right]}\beta_{t,m\prime }S\left(f_{dt},f_{st}\right)+n_{m_{*}^{\;}},$$
where
(26)$$D=diag\left\{e^{\left[j2\pi \alpha \left(m_{*}^{\;}-m\right)f_{s1}\right]},\ldots ,e^{\left[j2\pi \kappa \alpha \left(m_{*}^{\;}-m\right)f_{sN_{c}}\right]}\right\},$$
*diag*{∙} represents a diagonal matrix. Due to the fact that the degree of matrix *𝚽* is equal to the degree of matrix *𝚽D*, Equation (25) can be written as
(27)$$x_{m_{*}^{\prime }}=V_{m^{\prime }}\Xi_{m_{*}^{\prime }}+e^{\left[j2\pi \alpha \left(m_{*}^{\;}-m\right)f_{st}\right]}\beta_{t,m^{\prime }}S\left(f_{dt},f_{st}\right)+n_{m_{*}^{\;}},$$

From Equations (21) and (25) it is known that ***γ****_m*_* and ***γ****_m_* have the same clutter subspace and target signal model, which indicates the corresponding sparse structure of these vectors. Finally, the sparse echo data recovery model was established as
(28)X=ΨΥ+N, 
where
(29)$$\Upsilon =\left[\gamma_{1},\gamma_{2},\ldots ,\gamma_{M}\right],$$
(30)$$N=\left[n_{1},n_{2},\ldots ,n_{M}\right],$$
(31)$$X=\left[x_{1},x_{2},\ldots ,x_{M}\right],$$

## 4. Application of Sparse Recovery Algorithms

M-FOCCUS algorithm and OMP algorithm, which are typical joint sparse recovery algorithms, are used to solve the Equation (28) to estimate the spatio-temporal spectrum of clutter and target [19,20,21].

The estimation of *𝜰* with M-FOCCUS algorithm and OMP algorithm is equivalent to solving the following convex optimization problem [19]:(32)$$\min\|\Upsilon \;\|{}_{u,v},$$
(33)$$s.t.\;\|X-\Psi \Upsilon \;{\|}_{F}\leq \Sigma ,$$
where ‖*𝜰* ‖*_u,v_* = [‖ *𝜰*_1_*^T^* ‖*_u,v_* +…+ ‖ *𝜰**_r_^T^* ‖*_u,v_* …+ ‖ *𝜰**_NK_^T^* ‖*_u,v_*]^1/u^ denotes the *L_u,v_* norm of *𝜰*, *𝜰_r_* is r-th element of *𝜰*, *u* = 2, *v* ≤ 1. ‖∙ ‖*_F_* denotes the Frobenius norm of matrix and *Σ* is a constant determined by noise. The *L*_2,1_ norm of *𝜰* is the sum of the Euclidean norms of the columns of the matrix:(34)$$\|\Upsilon \;{\|}_{2,1}=\overset{m}{\underset{j=1}{\sum }}\|\gamma_{j}\;{\|}_{2},$$

The individual steps of the M-FOCCUS algorithm and OMP algorithm to solve Equation (28) were included in Appendix A and Appendix B, respectively.

## 5. Definition of Clutter Plus Noise Covariance Matrix and Weight Vector

As a result of determining the spectrum *𝜰* through a sparse recovery algorithm, the clutter plus noise covariance matrix (CNCM) ***Ř****_SR_* can be calculated by [19]
(35)$${\overset{ˇ}{R}}_{SR}=\sum_{p=1}^{K_{d}}\sum_{q=1}^{N_{s}}{\left|\gamma^{*}\left(p,q\right)\right|}^{2}S\left(f_{dp},f_{sq}\right)S^{H}\left(f_{dp},f_{sq}\right)+\sigma^{2}I_{NK},$$
(36)$$\left(p,q\right)\notin Ω\left(f_{st},f_{dt}\right),$$
where *p* = 1, 2, …, *K_d_*, *q* = 1, 2, …, *N_s_*, *𝜰*_0_^*^ is a column vector obtained by taking 2-norm of each row vector of *𝜰*_0_. *σ*^2^ denotes power of noise; ***I**_NK_* is a *NK* × *NK* identity matrix.

The possible Doppler frequency range of the target, which is determined by previously known information about the target is given as
(37)$$Ω\left(f_{st},f_{dt}\right)=\left\{\left(p,q\right)|\;\left|f_{dp}-f_{dt}\right|\leq \delta_{d}\;\&\;\left|f_{sq}-f_{st}\right|\leq \delta_{s}\right\},$$

System tolerances regarding Doppler frequency uncertainty and spatial frequency of the target are given as
(38)$$\delta_{d}=\mu_{d}{∆}_{d},$$
(39)$$\delta_{s}=\mu_{s}{∆}_{s},$$
∆*_d_* and ∆*_s_* are the resolution unit sizes specified by *K_d_* and *N_s_*. *μ_d_* and *μ_s_* are appropriate constants defined to prevent self-cancelling of the target.

If the CNCM has been determined from Equation (35), the optimal weight vector of the STAP processor can be determined by [19]
(40)$$w_{SR}={\overset{ˇ}{\mu R}}_{SR}^{-1}S\left(f_{dt},f_{st}\right),$$
where μ is the specified constant.

## 6. Simulation Results

The paragraph presents simulation results to show the effectiveness of the proposed STAP method. Simulation parameters are listed in Table 1, which refer to the standard parameters set in [20].

### 6.1. Performance of Spatio-Temporal Spectrum Estimation and Target Detection

First, the performance of the proposed method using the OMP algorithm is shown and analysed. The maximum number of iterations was 500. The units of Doppler and spatial frequency resolution are both equal to *N_s_* = *N_d_* = 60. The algorithm specifies *μ_s_* = *μ_d_* = 4. The following drawings are provided to confirm the performance of the proposed method for determining the CCM based on the OMP algorithm.

Figure 2 shows space-time spectrum of clutter before and after STAP processing in 2D charts. It shows the values of signals received by the MIMO radar array on the space-time plane. On the left chart, a yellow area is drawn along the diagonal of the graph. According to the literature on the subject of such research, this represents the so-called clutter ridge, whose graphic interpretation is shown in the Figure 3. In the right chart, it can be seen that the algorithm correctly detected the target located at the intersection of two straight lines, for a normalized Doppler frequency of *f_dt_* = 0.2 and a normalized spatial frequency also of *f_st_* = 0.2, respectively.

The following figure graphically depicts the interpretations of the clutter ridge. Clutter ridge is a constant value for given radar and environment parameters and depends directly on the speed of movement of the flying platform and inversely on the distance between the antennas and the pulses repetition frequency.

Figure 4 shows space-time spectrum of clutter before and after STAP processing in 3D chart. It is easy to see that clutter occurs in every distance cell. It is related to the movement of the flying platform and the non-zero value of the Doppler frequency shift between the platform and stationary field objects. The left chart shows the clutter ridge for simulation parameters. The proposed algorithm successfully removed the simulated clutter and enabled the detection of an object obscured by clutter.

A very important feature of the proposed STAP algorithm is the precise detection of objects. Another simulation was carried out for the same parameters. Figure 5 shows the values of signals received by the MIMO radar array as a function of range after the first pulse. At this stage, the received signals form a data cube of three dimensions (number of distance cells × number of pulses × number of antennas), which has not yet been processed by the newly developed STAP algorithm. Therefore, the radar cannot indicate the location of the object against the background of strong clutter. As you can easily see, the radar erroneously indicates that the object is 1000 m away from the radar.

Figure 6 shows the values of signals received by the MIMO radar array as a function of range after the first pulse. However, this time, the raw data were subjected to STAP processing by implementing the proposed STAP algorithm in the MATLAB environment. As you can easily see, the radar correctly indicates that the object is approximately 1900 m from the radar in a straight line.

### 6.2. Performance of Clutter Suppression

Next, we compare and analyse the proposed STAP method in terms of clutter suppression performance based on the improvement factor IF, where IF is defined as the signal-to-noise ratio at the output to the signal-to-noise at the input of STAP processor [21].

Figure 7 shows the performances of clutter suppression for both of the sparse recovery algorithms used. Considering the practical implementation and the standard parameter set [22], for the OMP algorithm, better clutter suppression can be obtained compared to the same STAP processing but using the MFOCUSS algorithm. This is due to the fact that the indentation of the IF curve in Figure 5 is narrower and reaches higher values. The use of the MFOCUSS algorithm in STAP processing and its comparison with the methods described in articles [12,13] were the subject of publication consideration [19].

## 7. Conclusions

The paper presents a new STAP processing method for target detection in a heterogeneous and non-stationary environment. The new method has been experimentally verified. The case of using MIMO radar on a flying platform was modelled and the OMP algorithm was used to determine the spatio-temporal clutter spectrum. The new method uses a single snapshot of the MIMO radar data cube radar. This allowed us to solve the problem of access to a large number of training cells and the non-stationary clutter in a heterogeneous environment, which in total significantly hinders the use of STAP processing in practice.

The paper alleges the analysis of the joint sparsity of echo data in the time and space domains in the MIMO on-board radar. Theoretical analysis and simulation results show that the proposed method can obtain a more accurate spatio-temporal spectrum estimation and have better clutter suppression performance than existing STAP methods using joint sparsity echo data and the MFOCUSS algorithm [12,13,19]. In addition, the OMP method is less computationally complex than the MFOCUSS method.

In summary, it has been proven in this paper that it is possible to estimate the STAP clutter covariance matrix by using the MIMO radar geometry model and OMP algorithm. The authors are aware of the lack of practical verification of the proposed algorithm; however, this will be targeted by future research.

## Figures and Tables

**Figure 1 sensors-20-03468-f001:**
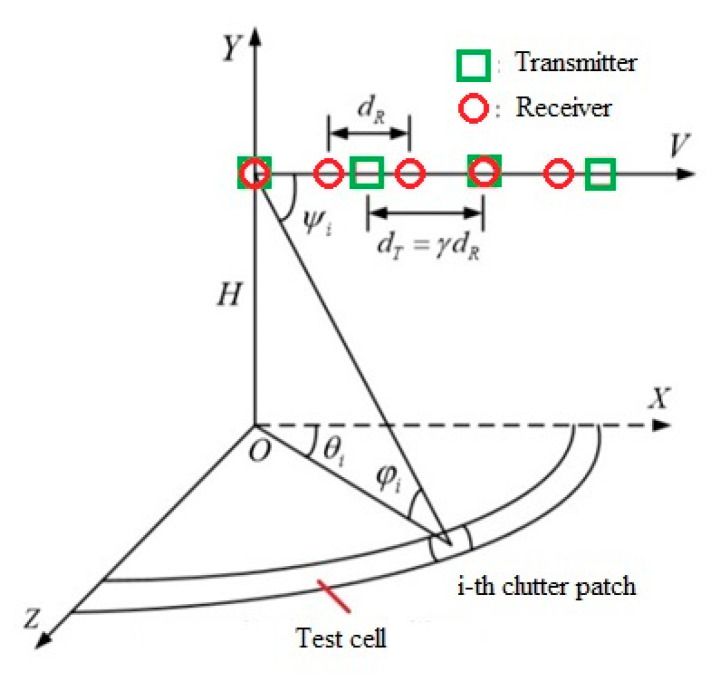
Geometry of the MIMO radar system under consideration.

**Figure 2 sensors-20-03468-f002:**
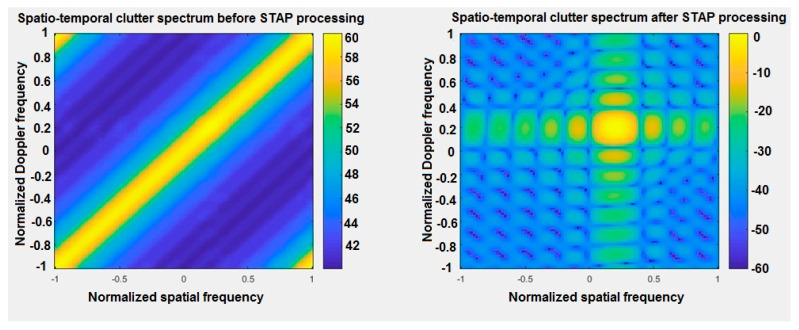
Space-time spectrum of clutter before and after STAP processing—2D charts.

**Figure 3 sensors-20-03468-f003:**
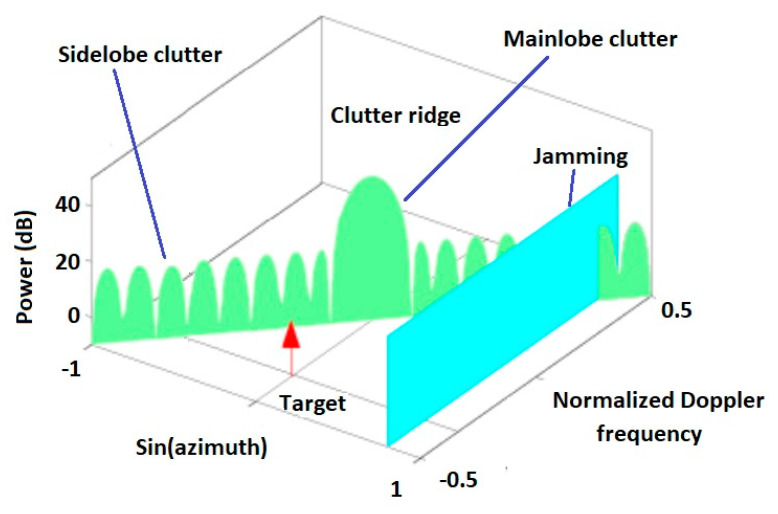
Clutter ridge in angular-Doppler plane.

**Figure 4 sensors-20-03468-f004:**
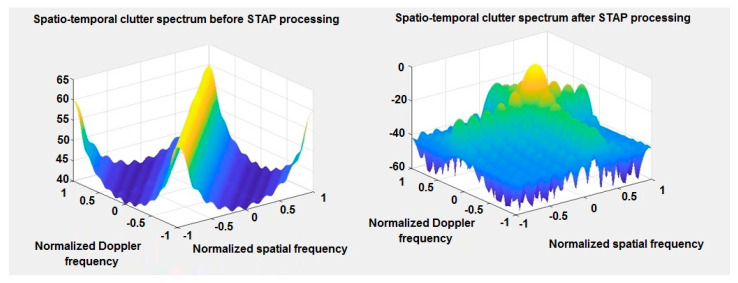
Space-time spectrum of clutter before and after STAP processing—3D charts.

**Figure 5 sensors-20-03468-f005:**
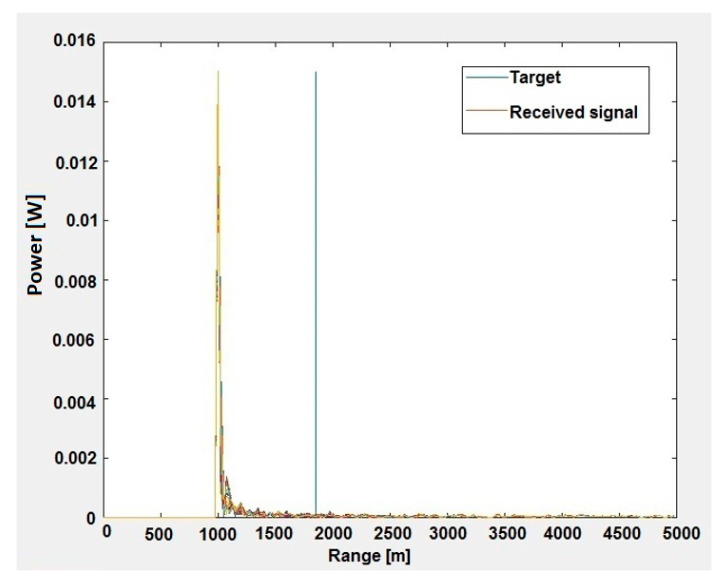
Values of signals received by the MIMO radar array as a function of range before STAP processing.

**Figure 6 sensors-20-03468-f006:**
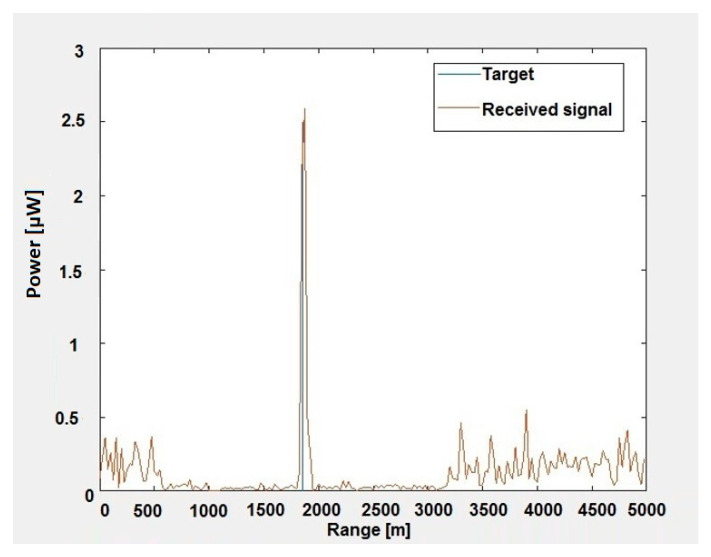
Values of signals received by the MIMO radar array as a function of range after STAP processing.

**Figure 7 sensors-20-03468-f007:**
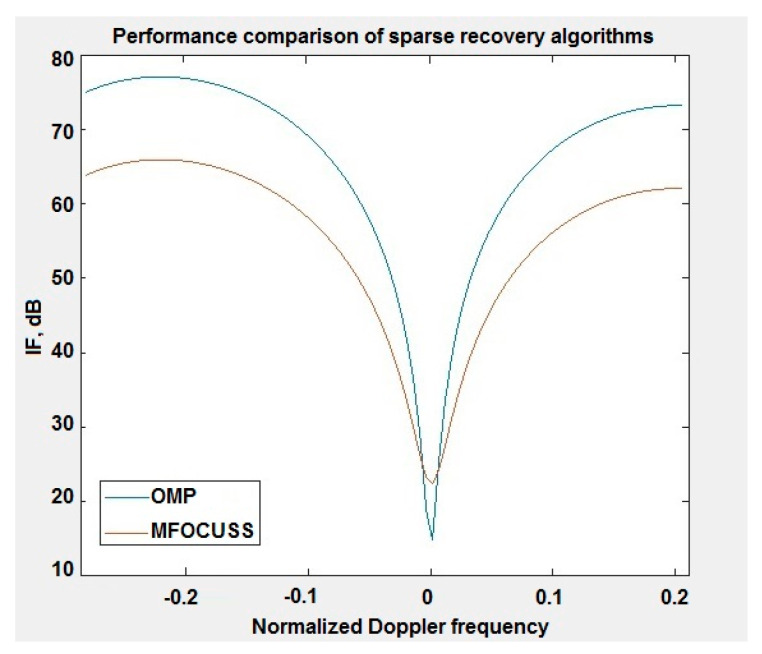
Performance comparison of sparse recovery algorithms.

**Table 1 sensors-20-03468-t001:** Parameters.

Parameter	Value
number of transmitters of MIMO radar	18
number of receivers of MIMO radar	8
number of pulses	8
wavelength	0.23 m
distance between transmitters	0.115 m
distance between receivers	0.115 m
distance between elements of the antenna array	0.115 m
flight altitude of the platform	5 km
velocity of the platform	250 m/s
pulse repetition frequency	4347.8 Hz
normalized Doppler frequency of target	0.2
normalized spatial frequency of target	0.2
clutter-to-noise ratio	30 dB
signal-to-noise ratio	10 dB

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
