# Peer review of "An Application of the Orthogonal Matching Pursuit Algorithm in Space-Time Adaptive Processing"

_sensors, 2020, doi:10.3390/s20123468_

Round 1

Reviewer 1 Report

[1] Grammar and writing style can be further improved.

[2] Figure 1: Please mark “theta_p”.

[3] Eqn.(1): Is f_R in the numerator or denominator?

[4] Line 65: Please elaborate “spatial frequency”.

[5] Eqn.(4): The last term should be indexed with “N_c” instead of “i”.

[6] Line 70: Is “alpha” the same as “gamma”?

[7] Line 96: Please briefly describe “theory of freedom of clutter”.

[8] Lines 100-104: Please rephrase these statements to improve readability.

[9] Line 113: Please elaborate why dealing with only the mth transmitted signal.

[10] Line 120: Please elaborate “L_{u,v} norm” and how is “v” involved.

[11] Line 128: Suggest to move the iteration index “t” to the superscript to avoid congestion.

[12] Section 6: Please provide more simulation cases to better validate the proposed algorithm.

Author Response

Thank you for your thorough review. The resulting corrections are included in the file.

Reviewer 2 Report

Congratulations on your well-founded and sophisticated research work on the area of targets detection. As I learned from your paper, the appropriate estimate for the CCM is a critical prerequisite for the application of the STAP algorithm. Here, you deduce in deep detail an mathematically well-founded approach to estimate the CCM. The topic of your paper is of high relevance for target detection and your approach is worth to be published.

However, I know that it is a demanding task to present extensive mathematical deductions in a paper. There hardly will be  many readers willing to follow all details of your deductions already in the main sections. A commonly chosen and more adequate way of presentation is to shift the algorithmic details to an appendix and to focus in the main sections just on the basic idea of the algorithm and the finally resulting approach.

In simulations, you verified the applicability and performance of you approach. However, the presentation of the simulation design and of the results is very condensed and too cursorily. You have to significantly increase this section. Let the reader know what are your considerations to choose just the parameters in the bullet list (e.g.: Do you base on the parameters of a given sensor? Are the parameters a tight test case for the estimation of the CCM?). By the way: A table would be a better readable way than a bullet list to present the values of the design parameters. You may find the plots in Figure 2-4 self-explanatory as you only cursorily discuss them. However, not all of your readers will be experts in target detection. Let the readers know, what features in the plots are relevant for the quality assessment and which conclusions you draw from the plots.

In your conclusions, I miss a conclusion beyond the trivial fact that your algorithm is proven (otherwise you had not decided to spend time in writing a paper on it). A conclusion is not a summary. Please tell the reader the lessons you learn from your research on this algorithm in more detail. You may also give an outlook where you see a need for future research work.

At last, I have a minor remark on the layout of your paper: How do you type the formulas and variables in your paper? The layout of the formulas is quiet poor and affects their readability. It looks like being typed as plain text. However depending on the text processing program you use, there is e.g. a formula editor in Word or a mathematical environment in LaTeX. It eases the readability when e.g. variable names are in another font as the surrounding text and when numerator and denominator of a fraction are stacked and do not occur in the same line, just separated by "/". What does the abbreviation "s.t."  mean, which occurs in eq. (19) and (28)? Is it a workaround for a mathematical symbol, you could not realize with the font, your used ?

To conclude my remarks: I recommend the publication of your approach. However, your paper shall be significantly restructured and it shall be completed w.r.t. simulations and conclusions. As I do not believe, that all of this work is applicable within the tight time constraints of MDPI for a minor revision, I see the paper in need for a major revision.

Author Response

(The authors gave the same response as above.)

Reviewer 3 Report

  1. Some variables are not denoted in the context.
  2. The sparse representation of clutter spectrum for STAP presented in this article is not new. The only difference between this manuscript and the existing literatures is that OMP algorithm is used to solve the optimization problem of sparse representation. But compared with MFOCUSS algorithm, OMP algorithm is a more basic method. Why the performance of OMP is better than that of MFOCUSS is not explained.
  3. In the simulation parameters, why is the distance between transmitters/receivers far less than wavelength?
  4. The IF curve in figure 4 seems to be incorrect. Generally, the horizontal axis of a IF curve represents the target Doppler frequency.
  5. The writing of the article is not rigorous enough. For example, references [8] and [12] are the same.

Author Response

(The authors gave the same response as above.)

Round 2

Reviewer 1 Report

[1] Figure 1, line 65: The description of “theta_p” in line 65 is inconsistent with that in Figure 1.

[2] Eqn.(6): If “alpha” is the same as “gamma”, suggest to use one of them instead of two different symbols.

Author Response

Thank you so much for the thorough review. I attach the source file with the corrections.

Reviewer 2 Report

Thank you for thoroughly considering my comments on the initial version of your paper. As I see, the quality of presentation is significantly improved.

Congratulations on managing such extensive changes in the text in such a short time.

In the current version, the paper is worth to be published as is.

Author Response

(The authors gave the same response as above.)

Reviewer 3 Report

The quality of simulation result figures should be improved.

Author Response

(The authors gave the same response as above.)
